# Changes in Certitude, Adherence, and Attitude: Immediate Effects of Rape Myth Intervention on Jurors in a Mock Trial

**DOI:** 10.3390/ijerph191610345

**Published:** 2022-08-19

**Authors:** Yazhi Pang, Kari Davies, Yong Liu

**Affiliations:** 1Department of Psychology and Human Development, Institution of Education, University College London, London WC1E 6BT, UK; 2Department of Psychology, Southwest University, Chongqing 400715, China; 3Department of Psychology, Bournemouth University, Poole BH12 5BB, UK

**Keywords:** rape myths, rape myth intervention, rape perceptions, judge’s direction, sexual violence

## Abstract

Previous studies have demonstrated the prevalence and negative consequences of rape myths in various social contexts, including their impact on jury decision-making. The current study adopted a mixed methods design to explore how educating jurors about rape myths via a judge’s direction affected their decision-making regarding the guilt or innocence of a defendant in a rape case. Data were obtained from two mock trials and 12 questionnaire responses. The sample consisted of 12 women participants aged from 20 to 25. The thematic analysis indicated that participants who received rape myths education exhibited resistance to rape myths, increased scrutiny of the defendant as opposed to the complainant, and less disbelief of the complainant’s alleged behaviours. Quantitative analysis suggested a strong positive correlation between the understanding of rape myths education and its influence on decision making; however, this was only found in the intervention group. Findings showed insights into the possible advantages of rape myths education for jurors that were delivered via the judge’s direction. Future research directions and implications were discussed.

## 1. Introduction

Attitudes are cultivated beliefs that can influence our perceptions with biased attitudes that prove harmful in many situations [1,2,3,4]. One of the most common biased attitudes is the belief in rape myths. Previous studies demonstrated that rape myths could alter subjective justice, which is the moral judgement of the behaviours in question based on personal blameworthiness instead of facts [5,6,7]. Rape myths are prevalent in various social contexts but are especially harmful when present in the criminal justice system; as such, attempts have been made by the criminal justice system to combat the effects of rape myths in court [8,9]. However, the existing policies provide no comprehensive solution to rape myth prevalence [8,10,11], and the underpinnings of a jury’s decision-making regarding rape cases remain unclear. The current study looked at how rape myths education delivered via a judge’s direction influenced the process of a juror’s decision-making. After providing definitions of rape myths and previous findings regarding the prevalence and the damage of rape myths, this introduction explored the influence rape myths have on the criminal justice system and existing interventions to combat them.

### 1.1. Rape Myths: Definition and Prevalence

Rape myths were first defined as “prejudicial, stereotyped or false beliefs about rape, rape victims, and rapist” [12] (p. 217). Rape myths have been used to “deny, downplay or justify sexually aggressive behaviours that men commit against women” [13] (p. 423). According to Bohner, rape myths fall into four main categories. The first category is beliefs that blame the victim, such as if someone dresses or acts in provocative ways, they are to blame if they are raped. The second category is beliefs that excuse the rapist, for instance, that once a man is sexually aroused, he cannot control himself. The third category is beliefs that doubt the allegation, such as women often lie about rape because they regret having sex. The fourth category is beliefs suggesting rape is exclusive to specific groups; for example, it is usually women who frequent bars who get raped [14].

The possession of rape myths can be found in many different populations cite. A study investigating the prevalence of rape myths in a mid-sized university found that out of 223 undergraduate participants, 22.4% of them agreed with certain rape myths [15]. Miller and Buddie found that 57% of 241 college students sampled endorsed some combinations of certain rape myths [10]. Using a sample of UK male university students, Hales and Gannon found that one in nine students reported committing sexual violence in the past two years, and aggressors were more likely to endorse offense-excusing myths about rape [16]. While most studies used student samples, a study by Basile using a non-student sample found a similar rape myth endorsement rate [17]. This can influence the legal processing of rape cases since the public comprises juries.

### 1.2. The Effects of Rape Myths on Survivors

Rape myths can impact how survivors perceive their experiences. One study, for instance, found that survivors of sexual assault might not acknowledge their non-consensual sexual encounters as assaults [18,19]. In a study of 296 participants, while 62.8% of them reported non-consensual sexual experiences, only 32.8% of them viewed their experiences as assaults. Researchers found the key components of whether the participants would identify their non-consensual sexual experiences as assaults were their acceptance of rape myths and attitudes towards sexual consent [18]. Similarly, Peterson and Muelenhard [20] found that participants who accepted rape myths and whose encounters corresponded with those myths were significantly less likely than other participants to acknowledge their experience as rape. It was also suggested this was a reason why some survivors felt guilty about their behaviours during the offence, such as blaming themselves for not physically resisting [20].

Rape myths also influence the recovery process of rape survivors. A study found that among various coping strategies, survivors who adopted avoidance coping recovered least successfully [21]. It was also found that rape myths substantially contribute to stigma and negative attitudes towards rape survivors. The stigmas and negative reactions that they received are key predictors of avoidance coping [22]. Moreover, survivors who labelled their experience as rape and reported a higher acceptance of rape myths consumed more alcohol per week and exhibited more depression symptoms compared to acknowledged survivors who reported lower acceptance of rape myths [22,23,24].

### 1.3. Rape Myths in the Criminal Justice System

From 2019 to 2020, the conviction rate for rape in England and Wales has hit an all-time low, with 55,130 rape reports resulting in only 1439 convictions, the equivalent of a 2.6% conviction rate [25]. One of the contributing factors to the low conviction rate is the impact of rape myths, which can be found at various stages of the criminal justice system. High rape myths’ acceptance, for instance, has been found among police forces [7], who are usually the first contact point when a rape survivor chooses to seek formal justice. Survivors are often asked stereotypical questions about the offence that correspond with rape myths, such as what they wore at the time of the offence, whether they physically resisted, or whether their encounter could have been perceived as consensual [11,26]. When police officers adhered to rape myths, they were more likely to discredit the survivors and cause them secondary victimisation, which might lead to the survivors becoming unwilling to continue with the investigation [27].

Extending to the later stages of the criminal justice system, rape myths also have an impact on jury trials. Two fundamental studies conducted in England identified three common functions of rape myths in trials. Rape myths were used to discredit, blame, and downplay the worthiness of justice protection for the complainant [5,28]. Survivors were judged as less trustworthy when they delayed reporting, had a lack of visible emotional distress, and of visible injuries [29,30,31,32,33]. Another study exploring the judges’ and barristers’ treatments of rape survivors in the context of the English and Welsh Criminal Justice Systems found that many of the hypothetical rationales used to assess the credibility of the survivors were based on rape myths. Delayed reports, lack of physical struggle, and other behaviours from the victim that were not aligned with common rape myths were considered irrational [20]. Regarding attributions of blame, research has also demonstrated that jurors who believe in rape myths were less likely to find the defendant guilty and place more accountability on the complainant [3,34]. A systematic review investigated mock juror studies across several countries, including the UK, the US, Spain, Italy, Germany, and Australia, and found both quantitative and qualitative evidence that higher acceptance of rape myths predicted victim blame [35]. A study by Sussenbach and colleagues observed that jurors who held a higher acceptance of rape myths focused their attention more on the complainant while jurors who held a lower acceptance of rape myths focused more of their attention on the defendant [36]; similarly, a study investigating English and Welsh rape trials found that jurors who held positive attitudes towards the defendant had more perceptions that the complainant contributed to the offence [37].

The research outlined above suggests there may exist a vicious cycle within the legal processing of rape allegations [7]. Performance evaluations and limited resources could lead to the police and prosecutors advancing cases that they anticipate holding a higher chance of conviction; this anticipation is derived from the assumption that stereotypes of rape influence jury decision-making [2]. Thus, this could cause the police and prosecutors to advance cases that correspond with rape myths, meaning that only those that adhere to a ‘rape myth stereotype’ are heard in court, judged by a jury, and convicted [7,38,39,40].

### 1.4. Interventions to Combat Rape Myths

Some effort has been made to combat rape myths, including the introduction of intervention programmes designed to lower rape myth acceptance. These programmes have been shown to be effective, although most are targeted at the public, and they do not investigate their effects when members of the public are called for jury duty. The common methods of delivery of these intervention programmes include courses, workshops, and videos [30]. Interventions usually include information on common rape myths, statistics that debunk the myths, sexual violence behaviours, gender equalities, and care for survivors of sexual assaults [41]. A study presenting rape myth education to a non-student sample, for example, was found to be effective. The education was delivered in video form; the control group watched a 10 min video on an irrelevant topic while the intervention group watched a 10 min video entailing rape myths education. After comparing their rape myth acceptance before and after the video exposure, the intervention group’s score after the video exposure significantly decreased compared to before video exposure and was significantly lower than that of the control group [42]. Another representative intervention programme, the SAIT (Sexual Assault Intervention Training) programme, consisting of one lecture on rape myths education, was found to significantly lower rape myth acceptance in its participants [43]. However, despite the effectiveness of existing interventions targeting the public, little evidence could be found regarding the immediate effects of rape myth interventions targeted specifically at jurors. The judge’s direction is a common procedure of English and Welsh courts, in which judges instruct and educate the jury on the facts or contexts that are relevant to the case in order to guide their deliberation [44]. In recent years, studies have started to investigate the effects of the judge’s direction on jury decision-making [21,45]. However, few of them have looked at their effects in rape case proceedings.

## 2. Methodology

### 2.1. The Present Study

Even though the prevalence of rape myths has been found to heavily interfere with the investigation of sexual assault cases [5,20,33,46], little investigation has been conducted on the effectiveness of rape myth interventions on jurors. To fill this gap, the current study adopted a mixed method design to test the immediate impact of rape myths education on juror’s decision-making through mock jury trials and the introduction of the judge’s directions. The current study aimed to investigate the influence of rape myths education on the jury’s decision-making. The study specifically aimed to answer (1) whether the rape myth intervention would influence participants’ perceptions of a rape case; (2) whether participants who did not receive the rape myth intervention would express more adherence to rape myths; and (3) whether the rape myth intervention would influence the judgement of guilt.

### 2.2. Participants

The study consisted of a total of 12 women participants aged from 20 to 25 (mean = 21.83, SD = 1.467). The criteria for participating were individuals over the age of 18 who were fluent in English. These criteria ensured that all participants had reached the legal age for becoming a juror in England and Wales and would possess no difficulty understanding the study on a linguistic level. All participants were also required to be eligible for UK jury duty, i.e., that they were not currently suffering from a mental disorder; were not on bail; had not been under detention or imprisonment for over five years, and had not been under a community order in the past 10 years [6]. The 12 participants were divided into two groups to represent two groups of jurors, with each group consisting of six participants.

### 2.3. Materials

The case chosen for the study was a real-life convicted case derived from the online legal information source LexisNexis.com. The written case record shown to the participants was a shortened version of the original report and contained details of the prosecution’s case, defendant’s case, and objective evidence regarding the incident. The written case record did not disclose the original verdict to avoid biasing the participants’ own deliberation (see Appendix A for these case materials).

The use of the judge’s direction is a procedural aspect of the English and Welsh court systems. The directions are set out and read by the judge to the jury conventionally before the deliberation; they include legal rules that jurors should follow to ensure a fair trial, such as the fact that the jurors ought to make their verdicts based on the evidence, that the prosecution would not need to prove the innocence of the defendant, but the guilt of the complainant [3]. The judge’s direction for the study was written by the researchers and then video-recorded by an actor acting as the judge. This actor was also a practicing barrister who was consulted in the creation of these materials. The standard judge’s direction in the study was like those given in real-life courts. The intervention group received the standard judge’s direction and rape myths education that cautioned jurors of three common rape myths and statistics about them.

The questionnaire was written on Qualtrics. The 10-item questionnaire collected the participants’ demographic information, including age, gender, and race. The questionnaire also asked the participants to report the group’s verdict and rate their agreeableness with the verdict, how easy they found it to understand the judge’s direction, influence of the judge’s direction on their decision-making, how much they thought the complainant was to be blamed, and how much they thought the defendant was to be blamed. Responses were given on a Likert scale of 1 to 5, with 5 being the highest or “most strongly agree” (see Appendix B for the questionnaire).

### 2.4. Procedure

The study received ethical approval on 6 February 2021, from the UCL Institute of Education Ethics Committee. The researchers started recruitment by posting the study on forums and social media. The information included the topic of the study, criteria for participating, time slots available for each group, and contact information. All participants voluntarily contacted the researchers for further details and were sent an information sheet and consent form via email. Participants were informed that the study was about decision-making in a rape case deliberation. No participants were aware of the investigation of rape myths education and the impact this may have on deliberation. For confidentiality purposes, each participant was assigned a unique identification number, which was attached to the consent forms. The participants then emailed the signed consent forms to the researchers to confirm their willingness to participate. The participants were assigned to either the control group or intervention group based on the time for which they were available; the participants were not aware of the difference between the control group and the intervention group.

The focus group meetings took place on Microsoft Teams. The researcher spent the first five minutes of the meeting introducing the study, which included the topic of the study, the presence of the researchers, their roles in the meeting, and what would happen during the meeting. The participants then received the written case record of a real-life rape case via email and were given 15 min to read through it. Participants were then shown the pre-recorded judge’s direction video via the researchers’ shared screen. The control group was shown the standard judge’s direction, while the intervention group was shown the standard judge’s direction with the added guidance on not adhering to rape myths or rape myths education. After going through all the materials, the researchers turned their cameras and microphones off, and participants deliberated on the case among themselves for a maximum of 45 min until they reached a verdict. Participants were informed that if they could not reach a decision after 45 min, they would be asked to vote on an outcome. Once the verdicts were made, the participants were sent questionnaires via email and were asked to fill them out. After their questionnaire responses were recorded, participants were then asked to return to the Teams meeting, where they were thanked and debriefed. After all the participants had completed the tasks, the focus group meetings were transcribed for analysis. The transcripts were first auto-generated by Microsoft Teams and then manually revised by the researchers.

### 2.5. Data Analysis

For the qualitative analysis of the transcripts, reflexive thematic analysis was conducted. The process of inductive coding began by repeatedly reading the transcript. Text that showed relevance to the study aims was identified. Thematic similarities and disparities were then noted and coded by the researcher. For example, “And him also lying, saying that he went straight home afterward, but the closed-circuit television (CCTV) did show he went to another building” and was coded as “comparison between offender’s account and objective evidence” and was later categorised under the theme “analysis of the evidence.”

For the quantitative analysis, Mann-Whitney U tests were performed to test for differences in variables between the control and intervention conditions. Spearman’s correlations were conducted to test for interactions between variables within each condition.

### 2.6. Reflexivity

Studies have indicated that women tend to possess less acceptance of rape myths than men [35]. As a woman researcher, there is a possibility that I am more inclined to data that fits my personal understanding of the issue compared to when data is analysed by a man researcher. As a feminist who is concerned with the issue of violence against women, I may be more sensitive to patterns that I view as an imbalanced power dynamic. Transcripts were repeatedly read while investigating from different perspectives, and the qualitative analysis was reviewed by co-authors to ensure less bias.

## 3. Results

### 3.1. Qualitative Analysis

Four themes were identified through the qualitative analysis: (1) analysis and interpretation of the evidence; (2) participants’ attitudes and trust; (3) rape stereotypes; (4) assessment of innocence or guilt.

### 3.2. Theme 1: Analysis and Interpretation of the Evidence

The first theme identified from the data was the jury’s observation of how the defendant’s and complainant’s accounts aligned (or not) from the objective evidence mentioned in the written case record, including the CCTV footage, inconclusive toxicology report of the complainant, and text messages and a video recording the complainant sent to her friend right after the incident. The two groups also made comparisons between and within the two accounts.

When given the same piece of evidence, the two groups reacted differently regarding the importance of the evidence they placed on their decision-making. Regarding the sobriety of the complainant at the time of the event, the toxicology report was inconclusive. Both groups exhibited doubts about the sobriety of the complainant. The control group made the following comments:

*It (the toxicology report) was inconclusive, so we don’t know her alcohol level at the time. Also, she said she was sober when she left. I mean, we do not know if she was actually sober when she left. I mean, I hate to say that, but we don’t have proof of that.* (Control Group 4)

*I was just trying to think, what types of drugs can make you lose your memory for a while.* (Control Group 4)

*And she is claiming she only had alcohol, again, we don’t know how true that is.* (Control Group 1)

The control group hinted at the possibility that the complainant took toxicants other than alcohol, and the other toxicants she might have taken could have influenced her memory and therefore influenced the truthfulness of her account. However, in the intervention group, the toxicology of the complainant did not seem to cause such overt suspicion and therefore did not seem to impair the credibility of the complainant. The intervention group held the following discussion regarding the defendant’s method of entering the complainant’s room:

*I mean it also says that she wasn’t drunk. She would have remembered, opening a door or. I mean it does say, that the toxicology was inconclusive. So, I mean, we don’t really know much about it.* (Intervention Group 5)

*Maybe like she was a bit drunk than she thought or, it’s really difficult to say.* (Intervention Group 1)

While the possibility that the complainant was intoxicated at the time of the event was pointed out by both groups, the intervention group did not seem to think this would have a major influence on the perceived credibility of the complainant. Participants from both groups were able to give logistical interpretations for both sides of the argument. However, how much value they place on the same piece of evidence was different, with less scrutiny being placed on the complainant by the intervention group when assessing her account.

Regarding this theme, when compared with the control group, the intervention group held little discussion over the toxicology report of the complainant. The lack of disputes hinted at a mutual trust in the complainant’s statement. It indicated that whether the complainant was drunk at the time of the event would not impact making the decision of guilt in the intervention group.

The analysis of this theme answered the research question that the two groups perceived the case differently in judging what evidence would have impacted their decision-making. Evidence that was deemed important in the control group did not matter as much in the intervention group. The intervention group also placed more scrutiny on the defendant compared with the complainant.

### 3.3. Theme 2: Participants’ Attitudes and Trust

The second theme that emerged from the data was the participants’ attitudes towards the defendant and complainant, as well as their respective trustworthiness. The attitudes participants held were shown to be largely reliant on previous themes, including the analysis of the evidence and their logistical reasonings.

Both groups exhibited similar negative attitudes towards the defendant, while the intervention group was slightly more explicit in these attitudes. According to the CCTV footage, the defendant went into another building after leaving the complainant’s building and put his hand through a window. Both groups came up with the possibility of the defendant causing harm to another individual. One participant from the control group said:

*The fact of putting the hand through window is sort of unclear. It’s just putting the hand through the window or if he was trying to harm somebody else?* (Control Group 2)

This comment also implicitly revealed this participant’s belief of the guilt of the defendant by pointing out the likelihood of the defendant harming another individual. The intervention group shared similar insights regarding the putting of the hand through the window. When discussing the method of entering the defendant into the complainant’s room, one participant from the intervention group made the following comment:

*Because they said when they saw him on the CCTV leaving, they saw him put his hands through another window. So, I was like maybe he did that to her as well? God knows, horrible. But he might have done that. Yeah, maybe he just broke in.* (Intervention Group 1)

While both groups suggested that the defendant could have harmed another individual, the intervention group used a more explicit word, “horrible,” exhibiting a stronger negative attitude towards the defendant.

The intervention group was more willing to express personal opinions of how they perceived the defendant compared with the control group. It was the defendant’s account that the complainant had initiated sexual intercourse and asked him to stop during the intercourse. One participant from the intervention group did not deem this statement as truthful when making the following comment:

*Also, I feel like he tried to portrait himself as respectful. And when he says she asked him to stop. Clearly if she told him to stop. I guess he didn’t stop.* (Intervention Group 3)

Stating that the defendant “tried to portrait himself as respectful” in a disbelieving manner, this participant suggests that she views the defendant in a negative way.

When analysing where the defendant’s and complainant’s accounts contradicted each other, both groups seemed to be much more trusting of the complainant’s account. Both groups used her account as evidence to disprove the defendant’s account. For instance, according to the defendant, the sexual encounter was initially consensual, and he stopped sexual activity when the complainant asked him to, while the complainant claimed that the defendant did not immediately stop. There was no corroborative evidence to support either account. When assessing this piece of information, the intervention group made the following comment:

*Also, I don’t feel like he stopped immediately because he said he repeated the words “You’re fine, you’re fine.” So clearly, he was not getting off of her right away.* (Intervention Group 2)

Comments exhibiting similar trust in the complainant’s account were also apparent in the control group. According to the complainant, she had never met the defendant prior to the incident, while the defendant claimed the opposite. Regarding this conflict of accounts, one participant from the control group made the following comment:

*She recounts the story where he told her, Uhm, “I’m Michael and we met at the club”, Obviously, I don’t know, we don’t have audio proof of that, but uhm, that would also go against him saying he’d met her in the corridor, but. I know technically you don’t have evidence.* (Control Group 2)

Analysis of this theme indicated both groups held a similar negative attitude towards the defendant. Compared with the control group, distrust in the defendant was expressed more explicitly by using strong negative wording such as “horrible”. This would indicate that the group who received the rape myth intervention were more vocal about their aversion towards the defendant. Analysis of this theme indicated that regardless of rape myth intervention, participants held relatively more negative attitudes toward the defendant compared to that towards the complainant. Both groups allocated much more trust and credit to the complainant. When the two accounts contradicted each other, both groups used the complainant’s account against the defendant’s account, thus deeming her account as more truthful.

### 3.4. Theme 3: Rape Stereotypes

The third theme that emerged from the data was how participants reacted to the case regarding details that did or did not fit common rape stereotypes. For instance, after expressing confusion about whether the defendant was intoxicated at the time of the event, one participant from the intervention group added the following comment:

*I mean it doesn’t justify what in case he did, but I was just wondering.* (Intervention Group 3)

By adding this comment, the participant made clear that she was aware of the possibility that alcohol and other intoxicants could be used to excuse the behaviour of the defendant (which is a belief held by many about rape) while simultaneously outlining that this should not be the case.

Another comment from the intervention group suggested a strong awareness of and resistance to rape myths. The intervention group expressed a lack of importance regarding the method of entrance in terms of the verdict by accentuating the lack of consent:

*In a way, it doesn’t matter if she brought him home or not, he still started to have sex with her when she didn’t want to have sex with him. So, it doesn’t change much at the end of the day.* (Intervention group 2)

This could be interpreted as a clear effort to combat the common rape myth that if a woman invited the man home, she is willing to have sex with him [47].

Conversely, in the control group, there was evidence of participants adhering to rape myths. One of the rape myths that was relevant to the control group was that survivors of sexual assaults act in a certain way, such as being visibly distressed [29,30,33]. When expressing their decision of the guilt of the defendant, one participant from the control group made the following comment about the complainant’s reaction to the incident:

*I think the man is guilty, but I just don’t know why the girl she, she thought she was raped, but then she after texting her friend she just went to sleep. Is it because she’s too tired or something like that?* (Control Group 4)

This comment showed that the participant found the complainant’s reaction to the incident unexpected, implying the existence of a “correct” reaction. The timing of the comment also showed importance here. Since the participant was about to decide the guilt or innocence of the defendant, making this specific comment at this time indicated that the answers to this question might impact their decision.

When another participant in the control group expressed their inclination to vote for guilty, they seemed to follow their decision by giving the defendant the benefit of the doubt, hinting at the possibility that his intentions might not have been to commit rape; this is related to the rape myth that men do not intend to force sex of women but can get carried away:

*You probably wouldn’t be able to do anything about intent or whatever, but based of the evidence, I feel like he is guilty. But then based off that, I guess, to some degree. Did he go there with the intention of doing stuff like that. I don’t know if he can get actual intent in these cases but, I feel guilty.* (Control Group 2)

Based on the analysis of this theme, the control group exhibited adherence to two common rape myths, first, that there should be a universal reaction to rape, such as crying and screaming, and second, that men can get carried away with their sexual urges while not intending to commit rape. The intervention group showed explicit resistance to the rape myth that alcohol and toxicants turn people into rapists and the rape myth that if a woman invited a man home, she is willing to have sex. The analysis of this theme answered the question that the control group expressed more adherence to rape myths while the intervention group made clear and conscious efforts to combat rape myths.

### 3.5. Theme 4: Assessment of Innocence or Guilt

The last theme derived from the data was how the two groups evaluated their decisions of guilty (both unanimously voted guilty) after analysing the evidence and deliberating. The two groups differed in how confident they were when expressing their decisions of guilty. When making the decision of guilty, both groups made cautious analysis of the existing evidence. However, while the intervention group showed more consensus on the verdict, with all participants agreeing with a guilty verdict, the control group’s decision of guilty was met with questions. The control group was less confident with the amount of evidence available for a solid verdict:

*I feel like there’s not that much evidence anyway, but… It doesn’t seem like he is telling the truth or he’s completely innocent.* (Control Group 1)

*I think he’s guilty, …, I don’t know if there is enough evidence.* (Control Group 1)

*I feel like we don’t have enough.* (Control Group 4)

*I feel like the problem is that we don’t have evidence for consent, we don’t know if she gave consent or not.* (Control Group 4)

*Compared with the control group, decisions of guilt in the intervention group were met with agreement: There are too many inconsistencies from the defendant’s side, so I will say guilty.* (Intervention Group 3)

*I agree with that.* (Intervention Group 5)

*I agree as well.* (Intervention Group 2)

*Especially the difference in his timing and her timing, it just doesn’t add up. And the CCTV in particular, that doesn’t lie…, I would say I am leaning towards guilty as well.* (Intervention group 1)

*I agree with the verdict.* (Intervention group 6)

Therefore, while both groups ultimately reached guilty verdicts, the consensus among the intervention group regarding the guilt of the defendant was more apparent, with all of them explicitly agreeing with the verdict.

There was less agreement among participants in the control group regarding the verdict, considering the repeated discussions over the sobriety of the complainant and the lack of adequate evidence. Even though both groups reached a unanimous decision, the intervention group was more assertive with their decisions. The analysis of this theme indicated that the jurors who received the rape myth intervention were more confident and certain of their decisions.

### 3.6. Quantitative Analyses

Based on the normality test of the data, Mann-Whitney U tests were performed to compare the differences regarding agreeableness with the verdict, the intelligibility of the judge’s direction, influence of the judge’s direction on the decision making, victim blame, and offender blame between the control group and the intervention group. No results were statistically significant (Table 1).

Spearman’s correlations were conducted to test for correlations between variables in each group (agreeableness with the verdict, intelligibility of the judge’s direction, the influence of the judge’s direction on the decision-making, victim blame, and offender blame). A strong positive correlation was found between the intelligibility of the judge’s direction and the influence of the judge’s direction in the intervention group, *r* = 0.853, *p* = 0.031 (Table 2). The judge’s direction for the intervention group included rape myths education, and was the only difference between the materials presented to the two groups. In other words, in the intervention group, the more the participants easily understood the judge’s direction, the more they reported it influenced their decision-making. The same trend was not found in the control group, indicating the judge’s direction did not significantly influence the control group’s decision-making (Table 3). However, due to the small sample size of the study, the quantitative results are likely not representative of the population as a whole and should be viewed with caution.

## 4. Discussion

The aim of the current study was to investigate whether rape myths education included in the judge’s directions would affect jury decision-making in a rape case. Thematic analysis revealed several themes and several differences in the two groups’ perceptions of the case. Both groups exhibited trust towards the complainant and negative attitudes toward the defendant, while the intervention group was more vocal about their distrust compared to the control group. The control group showed clear signs of adherence to rape myths while the intervention group made efforts to combat them, and the intervention group was more confident in their decisions of guilty compared with the control group.

Regarding perceptions of the case, even though both groups exhibited similar positive attitudes towards the complainant and negative attitudes towards the defendant, the intervention group placed more scrutiny on the defendant’s evidence and account compared with those on the complainant’s side. This corresponds with previous findings that there is an attention shift from the defendant to the complainant with increased rape myths’ acceptance [10,36]. The groups also differed in how they treated the same pieces of evidence. While presented with the same written case record, how much value the two groups placed on each piece of evidence was different. While the inconclusive toxicology report of the complainant did not seem to influence the decision-making of the intervention group, the control group viewed this piece of evidence as more impactful to their decision-making. Even though both groups exhibited negative attitudes towards the defendant, the intervention group was more explicit when expressing this attitude by directly using words with negative connotations. When expressing their decisions of guilt, the intervention group exhibited more certainty in the amount of evidence that was needed for a verdict, while the control group was hesitant.

Both groups exhibited more trust in the complainant compared with the defendant. In incidents where neither account was supported by objective evidence, both groups used the complainant’s account to disprove the defendant’s. However, since the toxicology report was inconclusive, the control group was more sceptical of the complainant’s truthfulness when she said she was sober during the incident.

The thematic analysis also revealed a strong effort to combat rape myths in the intervention group and more adherence to rape myths in the intervention group. Participants from the intervention group made explicit comments about the lack of consent in the case while also explicitly expressing that even if the defendant was intoxicated at the time of the event, this does not excuse rape. Conversely, participants from the control group made assumptions about the complainant’s behaviour according to certain rape myths and gave the defendant some degree of the benefit of the doubt.

Moreover, a positive correlation between how easily the jury understood the judge’s direction and how much influence this had on the decision-making was found in the intervention group, while such a correlation was not significant in the control group. Since the rape myth education was delivered through the judge’s direction, this correlation would indicate that the more participants understood the rape myths education, the more likely it would be to influence the decision they came to. No other correlations between the variables were found, and no statistical differences between the groups were found. As noted above, this could be due to the small sample size of the current study since a small sample size is more informative when using qualitative data rather than quantitative data [48], and further research would be required to more effectively explore these differences and correlations.

The current study showed support for previous research findings. Previous studies yielded results supporting the effectiveness of intervention programmes, and the current study also found that participants who received rape myths education showed a clear resistance to rape myths. While most previous interventions have been implemented for criminal justice system officials such as police officers and assessed for long-term results, the findings from the current study add to existing research by demonstrating the immediate effects of rape myths education on mock jurors. Previous studies have found that jurors used common rape myths as rationales to credit or discredit the complainant’s and the defendant’s alleged behaviours [21]. Similar situations were also present in the current study when one participant expressed the expectation of a more emotional reaction from the complainant. The participant’s expectation was based on rape stereotypes. The current study found that the intervention group placed more scrutiny on the defendant compared to on the complainant. Previous study has also found attentional shifts from the complainant to the defendant when jurors hold lower acceptance of rape myths [36].

The findings of the study have implications for future rape myth interventions in court. By demonstrating the influence rape myths education had on jurors, the findings suggested that including specific rape myths education during the judge’s directions may be helpful in generating less biased deliberations by jurors. Given that the delivery of the judge’s directions is a common procedure during almost all trials, this may be a practical and replicable intervention to improve court proceedings involving rape cases.

### Limitations and Future Research

One limitation of the study was the lack of gender variety as well as the volunteer bias within the sample. Participants were recruited by voluntarily contacting the researchers rather than the researchers contacting them. Out of all potential participants, only one identified as a man and was unable to attend because of a time conflict. While gender could be one of the mediating roles of rape myth interventions, it was not possible to assess this in the current study. Future studies could look at the mediating effects of gender on the awareness of sexual violence against women, as well as test for the immediate effectiveness of interventions with a gender-inclusive sample and by using other methods of sampling.

Another limitation of the study was that the power dynamics between the judge and jurors might have influenced the exhibited effectiveness of the rape myth intervention. Individuals are more inclined to conform when an authority figure has instructed them to do so [49]. Thus, participants in the current study might have exhibited signs of combating rape myths solely because the judge informed them about the incorrectness and unfairness of rape myths instead of truly changing their opinions. To explore the effect of the power dynamic on rape myth education, future research could investigate the immediate effect of rape myth interventions when it is delivered by a fellow juror who holds no power over other jurors.

Some of the procedural elements of the current study could be improved in future research. Due to the COVID-19 pandemic, the focus groups were conducted online. It was possible that holding the focus groups online affected the extent to which some participants engaged with the research. The researchers could not determine whether all participants were engaging with the materials and the discussions. Future research could be conducted in person, where researchers will be able to monitor the room and be aware of whether participants are engaging in the study.

## 5. Conclusions

By investigating the immediate influence of rape myth interventions through the delivery of the judge’s direction to potential jurors, the study suggested the effectiveness of rape myth interventions in combating rape myths. The study also revealed that rape myth interventions have the potential to influence the attitudes held towards the involved personnel in the rape case, the allocation of scrutiny and attention, and the trust placed in the defendant and the complainant. The findings further advocated for future interventions and research on education about the presence and influence of rape myths.

## Figures and Tables

**Table 1 ijerph-19-10345-t001:** Mann-Whitney U results with respect to groups.

	Agreeableness with the Verdict		Intelligibility of the Judge’s Direction		Influence of the Judge’s Direction on the Decision Making		Victim Blame		Offender Blame	
**Group**	M Rank	Z	M Rank	Z	M Rank	Z	M Rank	Z	M Rank	Z
**Control**	6.75	−0.27	7.50	−1.10	7.25	−0.76	6.5	0	7.00	−0.638
**Intervention**	6.25		5.50		5.75		6.5		6.00	

**Table 2 ijerph-19-10345-t002:** Correlation between the variables for the intervention group.

	Agreeableness with the Verdict	Intelligibility of the Judge’s Direction	Influence of the Judge’s Direction on the Decision Making	Victim Blame	Offender Blame
Agreeableness with the verdict			0.50	0.45	−0.71
Intelligibility of the judge’s direction			0.85 **	−0.32	−0.25
Influence of the judge’s direction on the decision making	0.50	0.85 **			−0.53
Victim blame	0.45	−0.32			−0.63
Offender blame	−0.71	−0.25	−0.53	−0.63	

Note: ** Correlation is significant at 0.05 (2-tailed).

**Table 3 ijerph-19-10345-t003:** Correlation coefficients between the variables for the control group.

	Agreeableness with the Verdict	Intelligibility of the Judge’s Direction	Influence of the Judge’s Direction on the Decision Making	Victim Blame	Offender Blame
Agreeableness with the verdict		−0.78	0.42	0.71	−0.28
Intelligibility of the judge’s direction	−0.78		−0.71	−0.63	−0.32
Influence of the judge’s direction on the decision making	0.42	−0.71		0.45	0.45
Victim blame	0.71	−0.63	0.45		0.20
Offender blame	−0.28	−0.32	0.45	0.20	

## Data Availability

The data presented in the study are available upon reasonable request from the corresponding authors.

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
