# Peer review of "Changes in Certitude, Adherence, and Attitude: Immediate Effects of Rape Myth Intervention on Jurors in a Mock Trial"

_ijerph, 2022, doi:10.3390/ijerph191610345_

Round 1

Reviewer 1 Report

Very relevant and impressive study, the topic is both, interesting to the readers of the journal and highly relevant. The impressive methodology, which really tackles the rape myths.

Some minor revisions should be made in order to improve the framing of the study and the reflexion of the limitations.

First, the nation state context ned more intensive framing, like the spreading of myths among students and police officers would definitely need country-specific context – the court system is however very crucial for the explanations and the contribution of the given paper, so the context of the country law must be given more intensively.

The pointing to the Gender division in both groups would be interesting. Some reflexion is important in the “limitation” part of the paper: More reflection on the role of the “judge” as presented to the study participants: how is the power relation here relevant (comparing to having anti-rape-Myths education done by a person, who is not in a “judge” position?) would be important. The participants of the study are always influenced by what they know about the study- so pointing to that and reflecting on this information would be important and should be added.

Reviewer 2 Report

I ask the authors to make the following modifications/corrections:

-The title "Methods" should be replaced with "Methodology" and moved to page 3 before the title "The Present Study".

-The first sentence from "discussions", from page 10 should be moved to "methodology" under the title "present study", page 4, after the word "directions" because it represents the purpose of the study.

-At the title "Procedure", page 4, in the first sentence, the number and date of approval from the Ethics Commission must be added.

-At the title "Procedure", page 5, the questionnaire with the 10 items must be added.

-On the title "Reflexivity", page 5, in the first sentence, "[Lewis, 2020]" must be replaced with the corresponding number from the bibliography.

- On page 7, in the last paragraph, sentence 4, a comma must be placed before the word "negatively".

- On page 9, under the heading "Quantitative analysis" the table with the study results should be placed.

-The conclusions must be rewritten. The conclusions must always refer to the results of the study. Bibliographic references are never included in the conclusions.

- On page 11, the number and date of approval from the Ethics Commission must be added.

- The bibliography must be corrected. Authors must comply with MDPI requirements when writing the bibliography (mdpi_references_guide_v5.pdf). Many references are incomplete or misspelled. There are many very old bibliographic references.

Thanks to the authors for their work!

Round 2

Reviewer 2 Report

Thanks to the authors for all the corrections made.

Good luck!